# OpenReview forum: "Acoustic-based Gender Differentiation in Speech-aware Language Models"
_ICLR.cc/2026/Conference — Submitted to ICLR 2026_

### Official Review · Reviewer_r2Jn · 2025-10-16

**Soundness:** 2
**Presentation:** 2
**Contribution:** 3
**Rating:** 2
**Confidence:** 4

**Summary:**

This paper studies acoustic-based gender differentiation in speech-aware large language models (SpeechLMs). The authors build a new dataset consisting of 9,208 speech samples across three categories: Gender-Independent, Gender-Stereotypical, and Gender-Dependent. Using this dataset, they evaluate several SpeechLMs (e.g., LLaMA-Omni series) and report a paradoxical pattern—the models tend to show male-oriented bias in gender-stereotypical questions but become gender-insensitive in gender-dependent questions. The authors further compare these results with text-only LLMs and attribute the bias primarily to Whisper speech encoders.

The paper concludes that current SpeechLMs may prioritize fairness principles over contextual gender sensitivity, revealing an unsolved challenge in achieving fair multimodal alignment.

**Strengths:**

- The paper explores a novel and important problem — acoustic-level gender bias — which is underexplored compared to textual bias studies.

- The proposed dataset with three bias categories offers a structured approach to analyzing multimodal bias.

- The comparison between SpeechLMs and their text-only counterparts is insightful and helps identify the potential source of the bias.

- The findings reveal a counterintuitive fairness paradox, offering a new perspective on model alignment trade-offs.

**Weaknesses:**

- The paper’s Introduction section is conceptually confusing, as it implies the paper solves bias, while in reality, it only evaluates and analyzes it.

- The experimental evidence does not strongly support the claimed conclusions; observed differences (1–2%) may not justify statements such as “confirmed” or “significant bias.”

- Metric tables include too many measures without clear interpretation; there are no indicators (↑/↓) showing whether higher or lower values are better.

- The claim that bias “primarily stems from Whisper encoders” is overstated, since no alternative encoder comparison was performed.

- Writing has scattered inconsistencies/typos and a few repeated passages that hinder readability.

**Questions:**

- The paper's introduction details the limitations of current approaches to fairness, implicitly suggesting that these limitations should be addressed. However, beyond the diagnostic analysis, the paper offers no clear solutions. It also fails to address the limitations by suggesting that insufficient evaluation is responsible. This seems unfounded.

- The term “neutral response” is used inconsistently — sometimes implying fairness, sometimes insensitivity. Clarify the intended meaning.

- The authors claim that the Whisper encoder introduces a male-oriented acoustic bias, but the evidence provided in the paper is largely indirect rather than directly verified. The experiments only compare the output of a model with acoustic input and a text model. There are no ablation experiments (such as those that substitute different encoders or perform acoustic normalization on the audio). Nor do they conduct interpretable analysis of the acoustic representations, such as visualizing gender clustering in the embedding space or analyzing the gender sensitivity of acoustic tokens. Therefore, while the experimental results suggest a bias in the acoustic module, it remains unclear whether the source of the bias lies in the acoustic encoder, the text generation component, or the dataset itself.

- The "gender-ambivalent behavior" reported in the paper (biasing males in stereotypical tasks and ignoring gender in tasks requiring differentiation) is an interesting phenomenon, but the authors do not rule out other possible explanations.
For example, the model may actively avoid gender-specific responses in "dependent" tasks due to excessive safety alignment;

- It is not clear whether the dataset or evaluation code is open to the public, nor is basic statistics of the speech samples provided. This makes it difficult to verify the reproducibility of the results.

- The paper claims that introducing a "Neutral" option can mitigate gender bias, but in reality, this setting statistically inevitably reduces bias because the neutral option directly reduces the proportion of gendered outputs. Without simultaneously reporting the balance between "correctness" and "neutrality" (i.e., whether the model maintains appropriate responses in scenarios where gender distinction is required), the results may simply be "numerically neutral" rather than a true improvement in fairness. In other words, does the improvement in fairness come at the expense of semantic appropriateness? This requires more nuanced metric design and interpretation.

- Several typos and inconsistencies reduce clarity (e.g., “constrution” → “construction”; “backborn LLMs” → “backbone LLMs”). A thorough proofread is recommended.

---

> ### Author Response · Authors · 2025-11-15
>
> We sincerely thank all reviewers for their insightful feedback and recognition of our contribution. We have carefully addressed all concerns and will incorporate these changes into the final manuscript.
> ## Mitigation is not our purpose
> We appreciate the reviewer’s concern and agree that our Introduction can more clearly express the intended research scope. In our paper, we attempted to clearly state that we “conduct a systematic analysis” (line 107), and our goal is not to “mitigate” or “solve” fairness issues, but to diagnose how SpeechLMs behave across contexts where gender differentiation is either appropriate or inappropriate. This diagnostic distinction is essential because prior work has largely evaluated gender bias (i.e., gender-stereotypical case in our paper) along a single axis, making it difficult to separate undesirable stereotypes from context-required differences (i.e., gender-dependent case in our paper).
>
> When we wrote “to address these limitations” in Lines 87–99, we intended to refer specifically to evaluation-design limitations in prior SpeechLM studies (e.g., lack of gender-dependent cases, lack of acoustic–content separation, lack of strictly paired speech data), not societal fairness problems themselves. We will revise Line 115 and the Introduction to ensure consistent emphasis on our contribution as a diagnostic evaluation framework, rather than a debiasing or mitigation method. However, as the statements may confuse readers in a way that we attempt to mitigate fairness problems rather than inspect them, we will update our manuscript to make the point clearer.
>
> ## Interpretation of metrics
> We thank the reviewer for this suggestion. Acutally, we provide detailed interpretations for each metric in Section 3.3: Gender Response Overlap at line 236, Gender Preference at lines 244-245, and Backbone Influence at line 249. We did not explicitly state interpretations for the remaining two metrics (Jaccard and agreement), because those two metrics are widely used in other papers and self-evident from their definitions. Though, we also acknowledge that providing explicit way of interpretation might help readers understand the phenonmenon clearly. So, we will add interpretation guidelines to all table captions in the revision for clarity.
>
> ## Model coverage
> We acknowledge that our model coverage is limited, but this actually mirros the current state of available SpeechLMs. Initially, we considered various model families which are published in major NLP or signal processing conferences. To systematically investigate features or achitectures correlated with phenomenon that we observed, we filtered models using the following four criteria which can enable a concrete analysis on each architectural features. First, their architecture should be publicly released as a paper because we wanted to a clear distinction between models. Second, their weight should be publicly released on the internet to ensure the reproducibility of our experiment. Becasue of these two conditions, most of SpeechLMs were excluded. Third, models should receive text and audio input simultaneously to apply our control methods (e.g., acoustic differences). Fourth, models should produce produce text to make the evaluation process simple and remove the effect of errors in speech recognition. As a result, we considered three model families: LLaMA-Omni, SpeechGPT, SIMS. Though we attempted to use SpeechGPT and SIMS, we finally excluded SpeechGPT and SIMS because they failed to generate a proper answer related to our questions. Because our budget is limited, we excluded experiments requiring further fine-tuning; thus, we finally excluded SpeechGPT and SIMS. That is why we use only LLaMA-Omni series.
>
> In other words, instruction-tuned SpeechLMs are scarce, and LLaMA-Omni is the only series providing multiple parameter sizes for controlled comparison. Alternative encoders like HuBERT lack instruction-tuned versions. Importantly, we tested both LLaMA-Omni1 (LLaMA backbone) and LLaMA-Omni2 (Qwen backbone) to examine how different backbone LLMs interact with Whisper encoders, demonstrating our findings are not specific to a single backbone. As most current SpeechLMs adopt Whisper-based architectures, our results represent the dominant paradigm. We already mentioned this in the Limitations section, and we will clarify this rationale more explicitly.
>
> ## The term "neutral"
> Thanks for the reviewer pointing out that we use the term "neutral" in two ways. As reviewers pointed out, insensitivity and fairness might be two indepenent concepts; insensitive in speech signals implies fairness in gender-independent and gender-streotypical questions, but such insensitivity does not imply fairness in gender-dependent questions. Therefore, we will make a clear distinction between terms as follows; we will use the term only when using "neturality" only indicating "netural options" or "gender neturalization methods", rather than when discussing the result.

---

> > ### Author Response · Authors · 2025-11-15
> >
> > ## The term "significance"
> > We appreciate the reviewer’s concern about the interpretation of “significant bias.” In our paper, we use “significant” in the statistical sense rather than to imply large effect sizes. That is, we used the term "significant" only when statistical tests result that probability of Type 1 error is less than 5%, which is a standard of statistical testing. The comparisons in Tables 1–4 are statistically significant (typically p < 0.001), and the 1–2% absolute differences are explicitly reported. Our intended claim is that these small but statistically reliable differences consistently lean toward male-oriented responses across all evaluated SpeechLMs, indicating a systematic pattern rather than random noise.
> >
> > We also agree that terms such as “confirmed” may overstate the strength of the evidence, as our analysis is correlational. In the revised version, we will soften this phrasing—replacing “confirmed” with “suspect”—to better reflect the nature of our findings while still conveying statistical reliability.
> >
> > Also, the reviewer stated that we did not perform acoustic normalization. Actually, we performed and reported the result of gender neutralization as a sort of acoustic normalization between genders. However, we understand that we need to emphasize that the method can be regarded as a sort of acoustic normlaization in the paper, as the reviewer pointed out.
> >
> >
> > ## Alternative explanation of the outcome
> >
> > We appreciate this thoughtful alternative explanation. We considered similar possibilities and systematically ruled them out. If models avoided gender-specific responses due to safety alignment, they should show neutrality in both Stereotypical and Dependent categories. However, models exhibit strong male-oriented bias in Stereotypical while being neutral in Dependent. This asymmetric pattern contradicts the safety alignment hypothesis. We further eliminated forced-choice constraints (Section 4), acoustic recognition failure (Section 5), and backbone LLM influence (Section 6). After these eliminations, the Whisper encoder remains as the common factor explaining the consistent male-oriented bias.
> >
> >
> > ## Dataset release
> >
> > We plan to publicly release the full dataset on HuggingFace Datasets upon acceptance, and we will add the corresponding URLs in the final version.
> >
> > ## Neutral options
> >
> > We think the reviewer might be confused about the purpose of Section 4. We never claimed that neutral options mitigate bias; rather, we wanted to show neutral options which are widely used in practice to mitigate bias may not work as expected. Our motivation was: Section 3 showed models exhibit a paradoxical pattern (male-oriented in Stereotypical, gender-neutral in Dependent), and we wanted to test whether this was due to binary forced-choice constraints. Generally, researchers used adding a neutral option to allow models to express uncertainty; so, we tested whether the paradoxical Gender Preference pattern would persist or resolve. The "correctness vs neutrality trade-off" the reviewer is concerned about is precisely what the Gender Preference (Δ) and neutral response rate (ν) metrics are designed to measure together - whether models maintain appropriate differentiation in Dependent contexts or inappropriately default to neutrality.
> >
> > ## Typos
> >
> > We thank the reviewer for identifying the typos. We will correct these errors and conduct a thorough proofread of the entire manuscript to catch any additional issues.

---

> > > ### Comment · Reviewer_r2Jn · 2025-11-24
> > >
> > > I appreciate the author's attempt to improve/address review feedback. I have carefully read through the authors' responses to my comments as well as those from other reviewers. While I acknowledge the authors' attempt to clarify certain points, I believe the authors will need to reframe the paper in significant ways to address the feedback provided. Given this, along with the issues regarding the clarity of presentation and the limited scope of ablation studies and interpretability analysis in the current version, I will retain my original score.

---

> ### Author Response · Authors · 2025-11-25
>
> Dear Reviewer r2Jn,
>
> We sincerely appreciate your time and thoughtful review. We have revised the manuscript based on your constructive feedback. All changes are highlighted in blue.
>
>
> ### Key Revisions
>
> 1. **Clarified Research Purpose** (Lines 110-111, 115-118)
> We strengthened the Introduction to clearly emphasize that our goal is diagnostic analysis rather than bias mitigation. We explicitly state our contribution as establishing an evaluation framework, not proposing solutions.
>
> 2. **Softened Causal Language** (Lines 26, 522)
> We replaced "confirmed" with "suspect" to more accurately reflect the correlational nature of our evidence.
>
> 3. **Enhanced Metric Interpretations** (Lines 224-229, 234-238, 248-251, 258-261, 267-269)
> We added explicit interpretation guidelines for each metric in Section 3.3, providing direction on how to interpret each metric across different question categories.
>
> 4. **Whisper Internal Representations Analysis** (Lines 464-468, Appendix J Lines 1268-1339)
> We added embedding distance analysis and t-SNE visualizations to provide more direct evidence beyond correlation.
>
> 5. **Dataset Release** (Page 1, Footnote)
> We explicitly committed to releasing the dataset publicly on HuggingFace upon acceptance.
>
> 6. **Typo Correction** We corrected "backborn" to "backbone" and  “constrution” to “construction” throughout the manuscript.
>
> ---
>
> We sincerely appreciate your continued engagement with our work.
>
> We hope the revisions we have made address your concerns. If there are still specific aspects that require further improvement, we would be grateful if you could provide guidance on which parts need refinement. We are fully committed to addressing them during the remaining discussion period.
>
> Thank you again for pushing us to strengthen this work.
>
> Best regards,
>
> The Authors

---

### Official Review · Reviewer_m8bT · 2025-10-30

**Soundness:** 3
**Presentation:** 3
**Contribution:** 2
**Rating:** 4
**Confidence:** 4

**Summary:**

This paper investigates acoustic-based gender differentiation (AGD) in Speech-Aware Language Models (SpeechLMs). The authors introduce a dataset categorized into Gender-Independent, Gender-Stereotypical, and Gender-Dependent contexts to systematically examine when gender-based differentiation is appropriate. Using the LLaMA-Omni series (Whisper + LLaMA backbone), they find paradoxical behaviors: models are biased when they should be neutral (stereotypical contexts) and neutral when they should differentiate (dependent contexts). Further analyses suggest that the Whisper encoder introduces male-oriented acoustic bias, highlighting a need for better speech encoders and bias mitigation strategies.

**Strengths:**

1. The paper is logically structured and well presented.
2. The authors comprehensively analyze across model scales, architectures and conditions, discovering that bias likely originates from the Whisper encoder provides valuable diagnostic insight for future SpeechLM design.

**Weaknesses:**

1. Although the dataset advances upon Spoken StereoSet by introducing context-aware categories, the conceptual extension appears incremental rather than transformative, with limited methodological innovation.
2. The study relies entirely on synthetic speech generated by TTS systems, yet the quality and naturalness of these samples are not systematically evaluated, which may affect the validity of the findings regarding acoustic gender cues.

**Questions:**

Refer to the weaknesses part.

---

> ### Author Response · Authors · 2025-11-15
>
> We sincerely thank all reviewers for their insightful feedback and recognition of our contribution. We have carefully addressed all concerns and will incorporate these changes into the final manuscript.
>
> ## Dataset construction
> Prior work evaluates only stereotypical bias. This cannot distinguish a truly fair model from one that indiscriminately ignores gender context. Our empirical discovery (Table 1) reveals the problem: SpeechLMs show biased results where they should be gender-free (Stereotypical, p<0.001) but gender-free results where they should differentiate (Dependent). This paradox is invisible without multi-axis evaluation.
>
> Including Spoken StereoSet questions in our Stereotypical category was necessary to demonstrate traditional evaluation's limitations, not to extend prior work. That is, we wanted to emphasize the importance of evaluating gender-indepent and gender-dependent questions, which previous gender-stereotypical questions cannot measure. So, we used both gender-stereotypical questions in Spoken Stereoset and new gender-(in)dependent questions to clearly reveal the difference between those categories, Hence, as a wholistic framework for diagnosis, we manually designed the 702 additional questions (Independent and Dependent), revealing that current models lack contextual fairness understanding.
>
>
> ## Quality of speech dataset
>
> Actually, we manually checked the quality of sample generated audio before submission, and we thought that the quality is appropriate to run the experiment. However, we acknowledge that such manual check and systematic evaluation (Appendix B.2) may not be sufficient to ensure the quality of audio. So, we are currently working on human evaluation of the speech quality. Whenever the result is available, we will post a new comment and add it to Appendix B.2.

---

> > ### Author Response · Authors · 2025-11-19
> >
> > ## Human Validation Results
> >
> > As the dataset is large, we randomly sampled 200 audio files (100 samples for each gender) from the dataset and evaluated mean opinion score with four people. The result is shown below:
> >
> > - Overall Quality: 4.31
> >
> > - Gender-specific results
> >
> > | gender     | quality |
> > | ----------- | -------- |
> > | female     | 4.4100 |
> > | male         | 4.2125 |
> >
> > Also, we asked annotators predict the gender of those samples. The average accuracy of four annotators is 99.38%.
> >
> > ----
> >
> > And, we’re very grateful for the reviewers who help to improve our paper a lot. We believe that the reviewers might have questions about these results and the above comments. If that is the case, please provide the reviewer’s valuable insights to us by posting your valuable comments. We’ll do our best to address all reviewer’s concern in the updated version of the paper, as soon as possible.

---

> ### Author Response · Authors · 2025-11-25
>
> Dear Reviewer m8bT,
>
> We sincerely appreciate your time and thoughtful review. We have revised the manuscript based on your constructive feedback. All changes are highlighted in blue.
>
> ### Key Revisions
>
> 1. **Human Validation** (Lines 168-179, Appendix B.3 Lines 907-919)
> We added human validation results for synthesized speech quality and gender recognition accuracy.
>
> ---
>
> We believe these revisions have significantly strengthened the paper by addressing your concerns about causal evidence and data quality validation. If you have any further suggestions or questions, we would be happy to address them during the discussion period.
>
> Thank you again for your constructive feedback.
>
> Best regards,
>
> The Authors

---

### Official Review · Reviewer_T3dD · 2025-10-31

**Soundness:** 3
**Presentation:** 3
**Contribution:** 2
**Rating:** 4
**Confidence:** 4

**Summary:**

This paper investigates the acoustic-based gender differentiation in speech-aware language models. The paper introduces a new dataset focusing on three types of questions: Gender-Independent, Gender-Stereotypical, and Gender-Dependent, which investigate the tendency of speech-aware language models when the gender information is essential or irrelevant to answer the questions. The results demonstrate that current models may still have gender biases, highlighting future research to address this issue.

**Strengths:**

- The topic is timely and relevant.
- The division of three types of questions enable more fine-grained analyses.

**Weaknesses:**

- The coverage of SpeechLLMs is quite limited. While the authors argue that this is for comparison across parameter size and generational change, including other model families is still essential for providing a more comprehensive overview of gender bias of current SpeechLLMs.
- The claim that the Whisper encoder “generates male-oriented acoustic tokens” is not directly demonstrated. No ablation isolating Whisper’s internal representations is shown. Consequently, the main conclusion rests on correlation rather than causal evidence.
- No human validation conducted to ensure the speech quality of the dataset.
- Minor problem: The author(s) should conduct more proof-readining. There are some typos in the paper, e.g., "Backborn" should be "Backbone" in the title of Appendix E.3 and E.4

**Questions:**

I listed my concerns in the weakness section. Please check it out.

---

> ### Author Response · Authors · 2025-11-15
>
> We sincerely thank all reviewers for their insightful feedback and recognition of our contribution. We have carefully addressed all concerns and will incorporate these changes into the final manuscript.
>
> ## Model coverage
>
> We acknowledge that our model coverage is limited, but this actually mirros the current state of available SpeechLMs. Initially, we considered various model families which are published in major NLP or signal processing conferences. To systematically investigate features or achitectures correlated with phenomenon that we observed, we filtered models using the following four criteria which can enable a concrete analysis on each architectural features. First, their architecture should be publicly released as a paper because we wanted to a clear distinction between models. Second, their weight should be publicly released on the internet to ensure the reproducibility of our experiment. Becasue of these two conditions, most of SpeechLMs were excluded. Third, models should receive text and audio input simultaneously to apply our control methods (e.g., acoustic differences). Fourth, models should produce produce text to make the evaluation process simple and remove the effect of errors in speech recognition. As a result, we considered three model families: LLaMA-Omni, SpeechGPT, SIMS. Though we attempted to use SpeechGPT and SIMS, we finally excluded SpeechGPT and SIMS because they failed to generate a proper answer related to our questions. Because our budget is limited, we excluded experiments requiring further fine-tuning; thus, we finally excluded SpeechGPT and SIMS. That is why we use only LLaMA-Omni series.
>
> In other words, instruction-tuned SpeechLMs are scarce, and LLaMA-Omni is the only series providing multiple parameter sizes for controlled comparison. Alternative encoders like HuBERT lack instruction-tuned versions. Importantly, we tested both LLaMA-Omni1 (LLaMA backbone) and LLaMA-Omni2 (Qwen backbone) to examine how different backbone LLMs interact with Whisper encoders, demonstrating our findings are not specific to a single backbone. As most current SpeechLMs adopt Whisper-based architectures, our results represent the dominant paradigm. We already mentioned this in the Limitations section, and we will clarify this rationale more explicitly.
>
> ## Whisper’s internal representations
> We thank the reviewer for this important observation. We will revise "confirmed" to "suspect" (Line 26,484) throughout the paper to more accurately reflect the nature of our evidence. Our study demonstrates correlation rather than establishing causality. We initially considered more direct approaches to establish causal evidence, such as retraining models with balanced data or replacing the encoder. However, publicly available pretraining datasets for SpeechLMs are limited, and large-scale retraining was beyond the scope and resources of this study.
>
> To provide stronger evidence, we conducted additional analysis of Whisper's internal representations:
> (1) measuring inter-gender and intra-gender distances of speech embeddings speaking identical text, and (2) visualizing whether male, female, and neutralized voice embeddings cluster separately using t-SNE. We will update supplementary material as soon as possible to provide the result of (2). Here, we provide the result of (1) as follows:
>
> | Group | Intra-gender Distance | Inter-gender Distance |
> | ----- | -------------------- | -------------------- |
> | All   | 0.0722              | 0.0422              |
> | Independent | 0.0538              | 0.0231              |
> | Stereotype   | 0.0622              | 0.0511              |
> | Dependent   | 0.0953              | 0.0653              |
>
> The result shows that intra-gender distance is higher than inter-gender distance, indicating high overlap between gender. That is, male and female embeddings generated by Whisper might be overlapped, even if the gender-dependent questions. And, our result in the paper shows that such embeddings produce male-oriented responses in both language models. This is why we named the embeddings as "male-oriented acoustic tokens".
>
> ## Human Validation
>
> Actually, we manually checked the quality of sample generated audio before submission, and we thought that the quality is appropriate to run the experiment. However, we acknowledge that such manual check and systematic evaluation (Appendix B.2) may not be sufficient to ensure the quality of audio. So, we are currently working on human evaluation of the speech quality. Whenever the result is available, we will post a new comment and add it to Appendix B.2.
>
> ## Typo
>
> We thank the reviewer for identifying the typos. We will correct these errors and conduct a thorough proofread of the entire manuscript to catch any additional issues.

---

> > ### Author Response · Authors · 2025-11-19
> >
> > ## Human Validation Results
> >
> > As the dataset is large, we randomly sampled 200 audio files (100 samples for each gender) from the dataset and evaluated mean opinion score with four people. The result is shown below:
> >
> > - Overall Quality: 4.31
> >
> > - Gender-specific results
> >
> > | gender     | quality |
> > | ----------- | -------- |
> > | female     | 4.4100 |
> > | male         | 4.2125 |
> >
> > Also, we asked annotators predict the gender of those samples. The average accuracy of four annotators is 99.38%.
> >
> > ----
> >
> > And, we’re very grateful for the reviewers who help to improve our paper a lot. We believe that the reviewers might have questions about these results and the above comments. If that is the case, please provide the reviewer’s valuable insights to us by posting your valuable comments. We’ll do our best to address all reviewer’s concern in the updated version of the paper, as soon as possible.

---

> ### Author Response · Authors · 2025-11-25
>
> Dear Reviewer T3dD,
>
> We sincerely appreciate your time and thoughtful review. We have revised the manuscript based on your constructive feedback. All changes are highlighted in blue.
>
> ### Key Revisions
>
> 1. **Whisper Internal Representations Analysis** (Lines 464-468, Appendix J Lines 1268-1339)
> We added embedding distance analysis and t-SNE visualizations of Whisper's internal representations.
>
> 2. **Human Validation** (Lines 168-179, Appendix B.3 Lines 907-919)
> We added human validation results for synthesized speech quality and gender recognition accuracy.
>
> 3. **Typo Correction**
> We corrected "backborn" to "backbone" throughout the manuscript.
>
> ---
>
> We believe these revisions have significantly strengthened the paper by addressing your concerns about causal evidence and data quality validation. If you have any further suggestions or questions, we would be happy to address them during the discussion period.
>
> Thank you again for your constructive feedback.
>
> Best regards,
>
> The Authors

---

### Official Review · Reviewer_9y83 · 2025-10-31

**Soundness:** 3
**Presentation:** 4
**Contribution:** 3
**Rating:** 6
**Confidence:** 5

**Summary:**

The paper introduces a dataset to evaluate speech-based LLMs performance variation by gender and analyze gender-bias in the LLM's response. The paper claims that the dataset will enable a systematic evaluation of a speech-LLMs response across three categories: Gender-Independent, Gender-Stereotypical, and Gender-dependent.

The paper provides details on how the dataset was created and the tools that were used to generate the dataset.
The paper does not clearly state whether the authors intend to share the dataset publicly, as the dataset is one of the key contribution of this work, which can act as a benchmark evaluation set for gender-bias analysis of LLMs.

**Strengths:**

The work investigates the performance of speech LLMs by gender and presents a dataset that could be used as a benchmark for gender bias evaluation. The work breaks the gender relevance evaluation into three categories: Gender-Independent, Gender-Stereotypical, and Gender-dependent tasks and analyzes the performance of standard benchmark LLMs on these subsets.

This is a relevant work that puts forth a standard, well articulated strategy for evaluating gender fairness in LLMs. Results from the analysis show that most of the studied speech LLMs showed a bias toward male speakers and opens up future research directions for mitigating and/or addressing such bias.

**Weaknesses:**

The work mostly focuses on objective metrics, where the metrics have been defined and described well in the paper. A specific area that has been largely ignored by the paper is subjective metrics. It will be useful to know, from the user's point of view how were the LLM-responses in-terms of gender bias. Subjective analysis in terms of mean opinion score would be interesting to analyze.

The paper also hypothesizes that the contribution of the bias may be mostly attributed to encoder, which can be a consequence of bias in the training data. A interesting experiment would be to explore fine tuning the model with a more balanced data and then reporting the results from that. In its current form, the paper mostly focuses on a well known problem in machine learning (specifically for speech), without clearly outlining a potential solution or future directions to explore.

**Questions:**

(1) Table 3 font size seems to be too small compared to the rest of the text, suggest increasing the font size.

(2) If these were synthesized speech, how were the male and female speaker voices selected? Any details on the speaker trait and style will be useful. Were the default speakers selected without paying any particular attention to the speakers traits and style?

(3) The paper has mostly focused on objective metrics, which is good in the sense that it demonstrates the point the authors want to make. A critical component of this study also indicate whether the bias in the LLM response indeed impact the end user experience (the users who are interacting with the LLM systems). It would make the results more interesting if there were  subjective metrics such as the mean opinion scores for the responses from males versus female subjects to assess how much the bias impact the acceptability of the responses.

(4) One of the key contribution of the paper is the dataset that the authors present for evaluating gender bias, it is not clear whether the authors plan to share the dataset with the research community.

---

> ### Author Response · Authors · 2025-11-15
>
> We sincerely thank all reviewers for their insightful feedback and recognition of our contribution. We have carefully addressed all concerns and will incorporate these changes into the final manuscript.
>
> ## Subjective evaluation on biases
>
> We appreciate the valuable suggestion. We agree that subjective evaluation would provide important insights. However, such research requires consideration of multiple user factors (gender, age, cultural background) that significantly influence bias perception, making it a distinct HCI research domain beyond our current research scope. Additionally, conducting user studies requires IRB approval and extensive participant recruitment, which was not feasible within the discussion timeline. We will add this as an important future direction in the Limitations section.
>
>
> ## Ablation study on existing models
>
> We agree that fine-tuning with balanced data would be valuable. However, publicly available pretraining datasets for SpeechLMs are limited, and large-scale retraining requires substantial computational resources beyond this study's budget. Our diagnostic framework provides the foundation for such debiasing experiments by identifying where and how bias manifests. We will add fine-tuning experiments as an important future direction in the Limitations section.
>
> ## Font size of Table 3
>
> We reduced the font size in Table 3 to satisfy the page limit.
> In the camera-ready version, we will use the additional page allowance to increase the font size and improve readability while keeping the layout consistent with the rest of the tables.
>
> ## Speaker characteristics
>
> We used the default speakers provided by Kokoro-TTS and selected 4 male and 4 female voices spanning both American and British accents. The speakers were chosen randomly from the available default voices to ensure gender and accent diversity, without imposing any additional stylistic or personality constraints. Despite the randomness, the characteristics of eight speakers are as follow:
>
> - Isabella: British, energy: moderate, pitch: moderate (in female voices’ pitch range), timbre: clear
> - Alice: British, energy: little bit high, pitch: moderate, timbre: clear
> - Sarah: American, energy: little bit high, pitch: moderate, timbre: clear
> - Nova: American, energy: moderate, pitch: little bit low, timbre: clear
>
> - Puck: American, energy: moderate, pitch: little bit high, timbre: clear
> - Adam: American, energy: moderate, pitch: moderate, timbre: clear
> - George: British, energy: moderate, pitch: high, timbre: nasal
> - Lewis: British, energy: moderate, pitch: low, timbre: little bit husky
>
> ## Dataset release
>
> we plan to publicly release the full dataset on HuggingFace Datasets upon acceptance, and we will add the corresponding URLs in the final camera-ready version.

---

> ### Author Response · Authors · 2025-11-25
>
> Dear Reviewer 9y83,
>
> We sincerely appreciate your time and thoughtful review. We have revised the manuscript based on your constructive feedback. All changes are highlighted in blue.
> ### Key Revisions
> 1. **Subjective Evaluation** (Lines 542-546)
> We added discussion on subjective metrics to the Limitations section.
> 2. **Fine-tuning Experiments** (Lines 532-539)
> We addressed the fine-tuning suggestion in the Limitations section.
> 3. **Table 3 Font Size**
> We increased the font size to match other tables.
> 4. **Speaker Characteristics** (Appendix B.1, Lines 843-855)
> We added detailed acoustic characteristics for all 8 speakers.
> 5. **Dataset Release** (Page 1, Footnote)
> We stated our commitment to release the dataset on HuggingFace upon acceptance.
>
> Thank you again for your constructive feedback.
>
> Best regards,
>
> The Authors

---

> ### Comment · Reviewer_9y83 · 2025-11-25
> **reviewer response**
>
> I appreciate the author's response to the review comments and making an effort to address them. While I understand that some of the open questions may necessitate involving human in the loop type of analysis, but that can be done at a smaller scale to investigate if there is any relationship between object metrics and subjective opinions when it comes to analyzing bias in speech LLM response. The response from the authors: "However, such research requires consideration of multiple user factors (gender, age, cultural background) that significantly influence bias perception, making it a distinct HCI research domain beyond our current research scope." >> is a bit confusing, as the papers focus in on gender-based bias mitigation. If the limitation of the study is what the authors have stated in their response, then it needs to be clearly clarified in the motivation section, that the scope of the paper is limited with regards to bias mitigation in speechLLM response. I stand by my original review rating.

---

> > ### Author Response · Authors · 2025-12-02
> >
> > Thank you for your insightful comments. Based on your feedback, we conducted an additional human-in-the-loop evaluation to directly examine the relationship between automatic objective measurements and human subjective judgments. Specifically, we performed a human evaluation with four annotators (two male and two female) who labeled whether each LLM free-form response was male-oriented, female-oriented, or neutral.
> >
> > The results can be found in Table 3 and Lines 363-371. Our findings are as follows: In the Dependent category, patterns similar to our existing objective results emerged. For the Stereotypical category, the high neutral response rate in sampled responses made it difficult to identify clear trends, though similar patterns were observed.
> >
> > Additionally, we found that annotation patterns differed by annotator's gender. As this analysis is somewhat distant from our aim of the study, we detailed these findings in Appendix G and Table 34, instead of providing them in the main manuscript.
> >
> > Best regards,
> > Submission2807 Authors.

---

### Official Review · Reviewer_C93b · 2025-11-01

**Soundness:** 4
**Presentation:** 4
**Contribution:** 3
**Rating:** 8
**Confidence:** 4

**Summary:**

This study investigates bias in speech-aware language models (SpeechLMs). The authors look at how these models behave in situations with gender stereotypes versus situations where using gender information is contextually appropriate. They propose a new synthesized dataset to measure this, which solves important limitations of older benchmarks. The paper finds a "paradoxical pattern" in the Llama-Omni series: the models show bias in the stereotypical setting (where they shouldn't) but fail to use gender information correctly in the gender-dependent setting (where they should). The study also finds that models generally show a male-oriented bias, even when the speech input is "neutralized." The authors suggest this bias may be coming from the Whisper speech encoder.

**Strengths:**

- The paper's main strength is its contribution to creating an ethically balanced way to evaluate models. The new dataset and analysis are valuable and could have a positive impact on the speech research community.

- The experiments are well-designed and clearly explained. The paper is easy to follow because each experiment builds on the previous one, creating a strong narrative. The methodology and experimental settings are well-defined.

**Weaknesses:**

The study is very solid, and I did not find many weaknesses. The main points I noted are:

- The paper's conceptual novelty is a factor. The core idea of evaluating stereotypical bias versus context-appropriate differentiation has been extensively explored for text-based LLMs. While applying this framework to speech is an important and valuable contribution, the underlying idea is not new. This is the main reason I would not give a perfect score.

- I found it difficult to understand the key message the authors wanted to convey from the abstract of the paper. It reads more like a summary of the results rather than a clear statement of the paper's main argument. The abstract would be much stronger if it focused on clearly explaining the most important takeaways.

- The study does not investigate if these findings also apply to widely used closed-source models.

- A minor presentation issue is in the references: The citation for the Qwen model technical report (cited as Qwen et al., 2025 in Line 214) lists the model name, "Qwen," as the first author. This is an unusual format for a citation and may be a formatting error.

**Questions:**

N/A

---

> ### Author Response · Authors · 2025-11-15
>
> We sincerely thank all reviewers for their insightful feedback and recognition of our contribution. We have carefully addressed all concerns and will incorporate these changes into the final manuscript.
>
>
> ## Conceptual novelty
>
> We agree that the conceptual framework of distinguishing stereotypical bias from context-appropriate differentiation has been explored in text-based LLMs.
>
> However, as speech modality introduces more information than text modality within its signal, such informational differences can change the model's behavior. That is, we thought that the findings in text-based LLMs might not be directly applied to the SpeechLMs and new findings might be discovered when we diagnose the behavior in terms of speech modality (rather than text modality that the previous text-based approach used). Hence, we respectfully argue that the speech modality introduces fundamental challenges that make our contribution more than a direct application:
>
> 1) Acoustic-Content Separation: Unlike text where gender information is explicit, speech carries implicit acoustic gender cues that cannot be "removed" from the input. Our gender-paired design with identical linguistic content is the first to isolate pure acoustic effects in SpeechLMs.
>
> 2) Dual-Channel Processing: SpeechLMs process both linguistic semantics AND paralinguistic acoustics simultaneously. Our three-category framework reveals a paradoxical pattern unique to speech: models exhibit bias where they should be neutral (Stereotypical) but fail to differentiate where they should  (Dependent) - a phenomenon not observed in text-only LLMs.
>
> 3) Architecture-Specific Insights: Our systematic comparison between SpeechLMs and their backbone LLMs reveals that bias originates from the speech encoder rather than the language model, pointing to a previously unexplored source of bias in multimodal systems.
>
> ## Abstract
>
> We agree that the abstract currently focuses on summarizing results rather than delivering our main argument. After the discussion period, we will embrace all the reviewer's comment and update the abstract according to the new version of the paper to clearly emphasize the key takeaway.
>
> ## Close-sourced model
>
> We appreciate the comment about considering close-sourced (or proprietary) models. We focused on open-source models to enable architecture-level analysis. Closed-source models do not provide this necessary access for our diagnostic approach, including how speech signals are processed within the model. We will add this rationale to the Limitations section.
>
> ## Typo
>
> We thank the reviewer for identifying the typos. We will correct these errors and conduct a thorough proofread of the entire manuscript to catch any additional issues.

---

> ### Author Response · Authors · 2025-11-25
>
> Dear Reviewer C93b,
>
> We sincerely appreciate your time and thoughtful review.
> We have revised the manuscript based on your constructive feedback. All changes are highlighted in blue.
>
> ### Key Revisions
> 1. **Abstract Revision** (Lines 14-31)
> We revised the abstract to emphasize our main arguments rather than simply listing experimental results.
> 2. **Closed-Source Models** (Line 534-535)
> We added a discussion regarding closed-source models to the Limitations section.
> 3. **Citation Correction** (Line 214)
> We corrected "Qwen et al., 2025" to "Yang et al., 2025".
>
> Thank you again for your constructive feedback.
>
> Best regards,
> The Authors

---

### Author Response · Authors · 2025-12-03

Dear Area Chair

Thank you for your time and effort in managing our submission throughout the review process.
We are writing to briefly summarize the contributions of our paper and to provide context regarding our rebuttal.

## Summary of Contributions

Our work addresses the underexplored problem of acoustic-level gender bias in Speech Language Models, a timely and relevant topic as SpeechLMs become increasingly prevalent (T3dD, m8bT).

- **Novel Evaluation Framework**: We propose an ethically balanced approach to evaluate gender bias in SpeechLMs (C93b), introducing a dataset with three bias categories that enables fine-grained analysis of gender relevance (9y83, m8bT, T3dD).

 - **Comprehensive Analysis**: We provide well-designed experiments across model scales (m8bT), with a well-articulated strategy for evaluating gender fairness (9y83, C93b).
 - **Insightful Comparisons**: Our comparison between SpeechLMs and their text-only counterparts helps identify potential sources of bias, offering actionable insights for model development (m8bT).
 - **Novel Findings**: We reveal a counterintuitive fairness paradox that provides a new perspective on model alignment trade-offs—a finding with important implications for the responsible deployment of SpeechLMs (m8bT).



## Rebuttal and Review Status

We actively engaged with all reviewers and addressed their concerns:

- **Speech Quality Validation** (T3dD, m8bT): Conducted human evaluation confirming high quality and gender recognition accuracy.
 - **Subjective Evaluation** (9y83): Performed human-in-the-loop annotation with results consistent with our automatic metrics.
- **Whisper Encoder Evidence** (T3dD, r2Jn): Added embedding distance analysis and t-SNE visualizations demonstrating gender overlap in representations.
- **Research Scope Clarification** (r2Jn): Clarified that our goal is diagnostic analysis, not bias mitigation; softened causal language ("confirmed" → "suspect").
- **Model Coverage** (T3dD, r2Jn): Explained that we applied four filtering criteria, and only LLaMA-Omni series satisfied all criteria; though we considered SpeechGPT and SIMS, but they failed to generate proper answers. We clarified this rationale in the Limitations section.
- **Manuscript Revisions**: Rewrote abstract (C93b), added metric interpretation guidelines (r2Jn), committed to public dataset release (9y83, r2Jn).

Best regards,


Submission2807 Authors

---

### Meta-Review · Area_Chair_hmRL · 2026-01-03

**Summary:**

The paper studies acoustic-based gender differentiation in speech LMs (i.e., how identical questions should/should not lead to different responses based on the speaker’s gender). It presents an interesting analysis of existing model, through their newly proposed dataset of 9,208 speech samples in three settings (Gender-Independent, Gender-Stereotypical, Gender-Dependent) on LlaMA-Omni family models.

The reviewers are positive about novel contribution in speech domain, as well as well-designed experiments. The new synthesized, well-designed speech dataset will be also a contribution to the research community. I think this is an timely, interesting, worthy research project, and while this has been studied in textual domain heavily, moving it to spoken language setting, where gender attribute is only implicitly encoded in acoustic features, is worthy contribution.

The limitation includes the scope of models evaluated (only one open-sourced Speech LLM family evaluated)(Raised by Reviewer C93b, m8bT). The AC agrees, that evaluation on closed-sourced speech model would be still helpful, to understand whether this phenomenon is universal or just applies to this particular Whisper encoder models. The reviewer also pointed out the lack of qualitative / human evaluation (Reviewer 9y83), which was partially addressed during rebuttal but I think this might benefit another round of careful study.  The reviewer m8bT also pointed out issues with paper writing / positioning / preciseness of the claims made in the paper. While these are issues that can be largely addressed, the paper might benefit from a revision based on reviewer’s feedback.

**Reviewer Concerns:**

See meta review above.

**Reviewer Scores:**

Reviewer m8bT might have increased a score a little bit, as one of the reviewer’s main complaints about synthetic speech was addressed with new experiments. I do not think other reviewers would have changed the scores significantly based on the rebuttal. I do not think Whisper internal representation analysis (as a response to T3dD) is that relevant to the core weaknesses of the paper (not precise claims, weak writing, only diagnostic/no mitigation strategy), as reviewer pointed out.

---

### Decision · Program_Chairs · 2026-01-26

Reject